

# Rapid global path planning algorithm for unmanned surface vehicles in large-scale and multi-island marine environments

Dong Wang[1,2], Jie Zhang[1,2], Jiucai Jin[2], Deqing Liu[2] and Xingpeng Mao[1]

[1] School of Electronics and Information Engineering, Harbin Institute of Technology, Harbin, Heilongjiang, China
[2] First Institute of Oceanography, Ministry of Natural Resources, Qingdao, Shandong, China

## ABSTRACT

A global path planning algorithm for unmanned surface vehicles (USVs) with short time requirements in large-scale and complex multi-island marine environments is proposed. The fast marching method-based path planning for USVs is performed on grid maps, resulting in a decrease in computer efficiency for larger maps. This can be mitigated by improving the algorithm process. In the proposed algorithm, path planning is performed twice in maps with different spatial resolution (SR) grids. The first path planning is performed in a low SR grid map to determine effective regions, and the second is executed in a high SR grid map to rapidly acquire the final high precision global path. In each path planning process, a modified inshore-distance-constraint fast marching square (IDC-FM$^2$) method is applied. Based on this method, the path portions around an obstacle can be constrained within a region determined by two inshore-distance parameters. The path planning results show that the proposed algorithm can generate smooth and safe global paths wherein the portions that bypass obstacles can be flexibly modified. Compared with the path planning based on the IDC-FM$^2$ method applied to a single grid map, this algorithm can significantly improve the calculation efficiency while maintaining the precision of the planned path.

# INTRODUCTION

Research on unmanned surface vehicles (USVs) has received increased attention in various military and civilian applications over recent years (*Yan et al., 2010*; *Campbell, Naeem & Irwin, 2012*; *Liu et al., 2016*). Robust and reliable navigation, guidance, and control (NGC) systems are required for USVs to perform a variety of complex marine missions. Path planning is an essential component of an NGC system. The main aim of path planning is to calculate a collision-free path with specific requirements. It determines the automation level of USVs and ensures the reliability and success of missions (*Tan et al., 2020*).

Global path planning is an important aspect of path planning. A variety of algorithms for the global path planning of USVs have been studied for different requirements. Dijkstra's algorithm (*Dijkstra, 1959*) is a classic graph-based algorithm that can plan the shortest global path for USVs (*Xie et al., 2016*; *Singh et al., 2018*). As an improvement of Dijkstra's

Corresponding authors
Jie Zhang, zhangjie@fio.org.cn
Jiucai Jin, jinjiucai@fio.org.cn

algorithm, the A* algorithm (*Hart, Nilsson & Raphael, 1968*) and its related improved algorithms are also commonly used in the global path planning of USVs (*Campbell, Naeem & Irwin, 2012*), such as the direction priority sequential selection method (*Naeem, Irwin & Yang, 2012*); the A* algorithm, which considers environment effects (EEA*) (*Lee et al., 2015*); and the angular rate-constrained Theta* (ARC-Theta*) algorithm, which considers both the angular rate and heading angle of USVs (*Kim et al., 2014*). These algorithms exhibit acceptable convergence and consistency; however, the planned paths must be further smoothed. The artificial potential field (APF) method (*Khatib, 1986*) is a classic method for both global and local path planning. The main drawback of the traditional APF method is the local minimum problem. Therefore, different studies (*Guo, Gao & Cui, 2013*; *Song, Hao & Su, 2020*) have been conducted to solve the local minimum problem for the path planning of robots or USVs. Evolutionary methods such as the genetic algorithm (GA) (*Kim et al., 2017*; *Arzamendia et al., 2019*; *Wang et al., 2020*), particle swarm optimization (PSO) (*Song et al., 2015*), and ant colony optimization (ACO) (*Song, 2014*; *Xia et al., 2019*) have also been applied in the path planning of USVs. GA is robust and adaptable, but it has the shortcomings of poor local search ability and the premature convergence phenomenon. Therefore, other methods, such as the simulated annealing algorithm, are usually introduced to improve the performance of GA (*Zhang, Xu & Xie, 2019*). Nevertheless, the path planned by GA still lacks consistency. Similar to GA, PSO has the advantage of strong robustness and the shortcoming of premature convergence. However, it is dependent on various parameters, and there is no specific theory for guiding the setting of these parameters for different problems. Similarly, the parameter selection of ACO is more dependent on experience, and this algorithm can easily fall into local extremum.

In some special applications, the shortest time may be an important requirement for USVs. The fast marching method (FMM) can be a solution for time-optimal global path planning. This method was first proposed by *Tsitsiklis (1995)*, and *Adalsteinsson & Sethian (1995)* independently and was extended by *Sethian (1999)*. An experimental survey for nine different FMMs is shown in *Gómez et al. (2019)*. The path planned by the FMM is usually extremely close to the obstacles. One solution is to adjust the speed map, as exemplified by the method with an adjusted cost function (*Messias et al., 2014*) and the FM$^2$ method (*Garrido et al., 2007*). FMM-based methods have been widely used in path planning applications (*Gómez et al., 2013*; *Amorim & Ventura, 2014*; *Alvarez et al., 2015*; *González et al., 2016*). Marine applications based on FMM were introduced by *Garrido, Alvarez & Moreno (2020)*. Interesting modifications for the FM$^2$ method have been performed, and the FMM has been subjected to a vector field considering the effects of several vector variables such as wind flow or water currents (*Garrido, Alvarez & Moreno, 2020*). In addition, studies on path following and obstacle avoidance and formations have also been conducted using FMM (*Garrido, Alvarez & Moreno, 2020*). USV formation path planning has also been performed by *Liu & Bucknall (2015)* and *Tan et al. (2020)*, and an angle-guidance FM$^2$ method has been used for the Springer USV to make the generated path compliant with the dynamics and orientation restrictions of USVs (*Liu & Bucknall, 2016*; *Liu, Bucknall & Zhang, 2017*). An improved anisotropic fast marching method using a

multi-layered fast marching was proposed by *Song, Liu & Bucknall (2017)*, which combines different environmental factors and provides interesting results. In addition to the global path planning, the FMM-based methods also show potential in collision avoidance of USVs (*Wang, Jin & Er, 2019*; *Garrido, Alvarez & Moreno, 2020*). These successful studies have demonstrated the potential of FMM-based methods in global path planning of USVs.

The basic FMM can plan time-optimal paths for USVs. However, some of the main shortcomings are that the paths planned by the FMM are too close to obstacles, and there may be abrupt turns when the paths bypass obstacles with sharp corners. Thus, the FM$^2$ method is proposed to address these problems, and two dimensionless parameters are introduced to adjust the paths more flexibly (*Garrido et al., 2007*; *Garrido, Alvarez & Moreno, 2020*). However, the two introduced parameters lack clear physical meaning, and the suitable adjustment is difficult to identify with specific values. Moreover, the adjustment effects of these two parameters are not common because the adjustment degree with the same parameters varies with different grid maps. Another common problem about FM-based methods is that the computational efficiency of path planning decreases sharply when the scale of the grid map is very large. Therefore, we make two improvements to address the mentioned shortcomings of the basic FM$^2$ method. First, we introduced an inshore-distance-constraint fast marching square (IDC-FM$^2$) method to improve the inshore path adjustment performance for the first time. Comparing with the basic FM$^2$ method, the IDC-FM$^2$ method applies two inshore distance parameters other than the two dimensionless parameters to adjust the paths around the obstacles. The adjustment effects for path planning of the IDC-FM$^2$ method are stable in different grid maps as the IDC-FM$^2$ method can constrain the path portions around the obstacles within the region constrained by the two inshore distance parameters. Further, to improve the computational efficiency, the algorithm that applies the IDC-FM$^2$ method based on two-level spatial resolution grid maps is designed.

## RELATED METHODS

### Environment map model

In 2D global path planning applications based on the FMM or its improved methods, discrete numerical calculations are based on cartesian grid maps. Therefore, an environment map should first be converted into a binary grid map with a suitable spatial resolution (SR). Free and open-source satellite images (such as Google satellite images) can be used as the data source for the maps in most USV applications, and the corresponding grid maps can be generated using image processing (*Shi et al., 2018*) combined with manual assistance. The binary cell is set as an obstacle cell when an obstacle exists at the geographic location (0 values); otherwise, it is set as a free cell (1 value).

### Fast marching square method

When using FM-based methods to plan a path, the basis is the FMM. The core work of the FMM is calculating solutions of the Eikonal equation (*Tsitsiklis, 1995*; *Adalsteinsson & Sethian, 1995*). The Eikonal equation describes a wave front propagation scenario from sources with the speed of a wave front given as $F_x$ at cell $x$. It can be expressed as

$\|\nabla T_x\| F_x = 1$, where $T_x$ is the arrival time of the wave from the source to the cell $x$ and $\nabla$ is a vector differential operator. From the perspective of time-cost, the Eikonal equation can be expressed in another form (that of *Lin, 2003*):

$$\|\nabla T_x\| = \tau_x \qquad (1)$$

where $\tau_x$ is the time-cost at cell $x$ and is equivalent to $1/F_x$. The solution we want to calculate is $T_x$, and all of them compose the arrival time map, $\mathcal{T}(x)$. All time-cost $\tau_x$ compose the time-cost function map, $\boldsymbol{\tau}(x)$.

The FMM was first proposed independently by *Tsitsiklis (1995)* and *Adalsteinsson & Sethian (1995)*.The solution $T_x$ in cell $x$ can be interpreted as the wave arrival time from the nearest source to cell $x$. Three cell sets (the accepted set $\mathcal{S}_A$, trial set $\mathcal{S}_T$, and far set $\mathcal{S}_F$) are defined to calculate $\mathcal{T}(x)$. The set of all cells at which $T_x$ will not change is $\mathcal{S}_A$. $\mathcal{S}_T$ is the set of cells to be examined. Every $T_x$ in $\mathcal{S}_T$ has been previously computed but may be updated. $\mathcal{S}_F$ is the set of all other cells with $T_x$, which has never been computed (*Lin, 2003*).

The calculation process includes initialization and loop procedures. During initialization, source cells $x_0$ with $T_{x_0} = 0$ are set to $\mathcal{S}_T$, and all other cells are set to $\mathcal{S}_F$ with $T_x = \infty$. After the initialization, the cell $x_m$ with the smallest $T$ value in $\mathcal{S}_T$ is selected and moved from $\mathcal{S}_T$ to $\mathcal{S}_A$. Thereafter, the non-accepted neighbor cells of $x_m$ are updated, including the solutions and cell sets. Considering Tsitsiklis's deduction (*Tsitsiklis, 1995*; *Lin, 2003*), each of the new neighbor solutions is determined by:

$$T_x = \min\left(\tilde{T}_x, \left(T_{x_m x_i}, i = 1, 2\right)\right) \qquad (2)$$

where $\tilde{T}_x$ denotes the original solution. $T_{x_m x_i}, i = 1, 2$ are the candidate solutions for the paths that pass through the line segment $x_m x_i$ and propagate to cell $x$ (see Fig. 1), which are calculated by:

$$T_{x_m x_i} = \begin{cases} \frac{1}{2}\left(T_{x_m} + T_{x_i} + \sqrt{2\tau_x^2 - \left(T_{x_m} - T_{x_i}\right)^2}\right), & T_{x_m x_i} > T_{x_m}, \text{and } T_{x_m x_i} > T_{x_i} \\ \min\left(T_{x_m}, T_{x_i}\right) + \tau_x, & \text{others} \end{cases} \qquad (3)$$

where $T_{x_m}$ and $T_{x_i}$ are the accepted solutions at cells $x_m$ and $x_i$, respectively. If cell $x_i$ is in $\mathcal{S}_T$, $T_{x_i}$ is $\infty$. For the cell set, if a non-accepted neighbor cell $x$ is in $\mathcal{S}_F$, it moves from $\mathcal{S}_F$ to $\mathcal{S}_T$. The loop procedure is performed continually until all cells are in $\mathcal{S}_A$. The resulting map is the arrival time map, $\mathcal{T}$.

The main disadvantage of the basic FMM is that the computed path is too close to obstacles and forces the vehicle to perform abrupt turns (*Garrido, Alvarez & Moreno, 2020*). As described in *Garrido et al. (2007)* and *Garrido, Alvarez & Moreno (2020)*, a smooth path with sufficient safety distances from obstacles can be computed using the FM$^2$ method. This method applies the basic FMM twice. The procedure for computing paths is described as follows (*Garrido, Alvarez & Moreno, 2020*):

1. The environment is modeled as a binary grid map (Fig. 2A).
2. The FM-1st step. All obstacle cells are used as wave sources ($T = 0$), expanding several waves at the same time at a constant speed. The value of each cell in the resulting map indicates the time required for a wave to reach the closest obstacle (see Fig. 2B). This

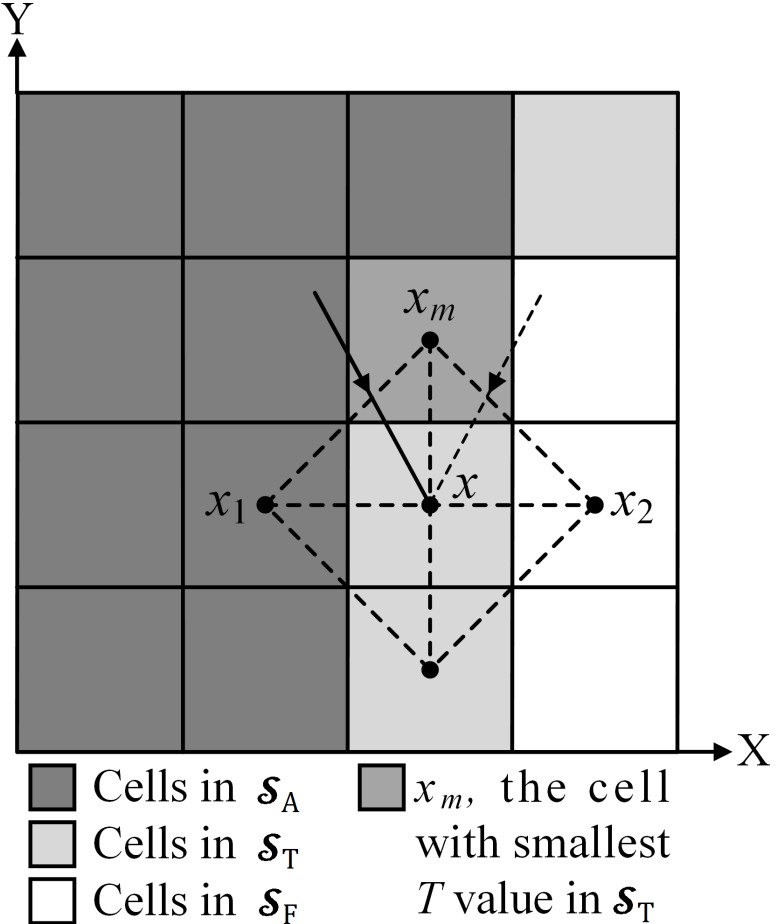

Cells in $\mathbf{S}_\mathrm{A}$
Cells in $\mathbf{S}_\mathrm{T}$
Cells in $\mathbf{S}_\mathrm{F}$

$x_m$, the cell with smallest $T$ value in $\mathbf{S}_\mathrm{T}$

**Figure 1** **Diagram of updating the neighbor of $x_m$ in a 4-neighbor scheme.** Cell $x$ is the neighbor cell to be updated. In this case, $x$ is in $\mathbf{S}_T$. If $x$ is a cell in $\mathbf{S}_F$, it moves from $\mathbf{S}_F$ to $\mathbf{S}_T$.

is proportional to the distance from the obstacles. By reversing the meaning of these values, they can be understood as the maximum admissible speed at each cell. Finally, the speed values are rescaled to fix a maximum cell value of 1.

Modifications of the speed map with two parameters, $\alpha$ and $\beta$, can adjust distances between the computed paths and obstacles. The value of each cell in the speed map $F_{i,j}$ is adjusted exponentially by $\alpha$:

$$newF_{i,j} = F_{i,j}^\alpha \tag{4}$$

The parameter $\beta$ is used to saturate the values in the speed map. It is defined within the range of 0 and 1. Every $F_{i,j}$ with a value greater than $\beta$ is set to one (see Fig. 2C).

Comparing with a path without modifications, the path modified by $\alpha$ will be closer to obstacles when $\alpha < 1$. On the contrary, if $\alpha > 1$, the modified path will stay further away from obstacles. The parameter $\beta$ allows the path to move closer to obstacles. When $\beta$ is smaller, the modified path will be closer to obstacles.

3. The FM-2nd step. The goal point is used as a unique wave source. The wave is expanded over the map until the starting point is reached. The speed at each cell (equivalent to

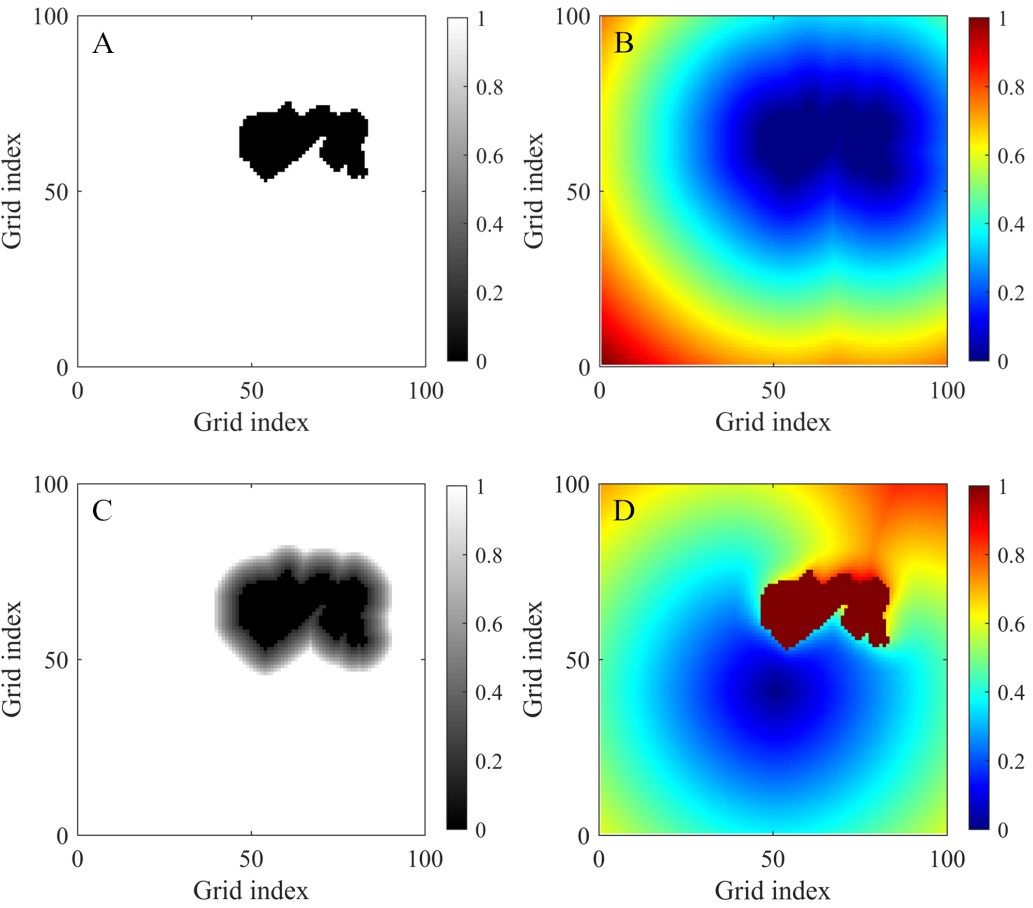

**Figure 2** **Maps used or generated in the process of using the FM² method.** (A) Binary grid map of an obstacle environment. The black cells are obstacle cells, and the white cells are free cells. (B) Arrival time map expanding from the obstacles, which is proportional to the distance from the obstacles. (C) Speed map with a saturated value. The black areas (0 value) represent the obstacles, while the white areas (1 value) represent the areas with velocity saturation. The other values are rescaled to better demonstrate the trend. (D) Arrival time map obtained by applying the FM² method with $x_g = (50, 40)$. The path can be acquired by using the gradient descent method based on this map.

$1/\tau_x$) is obtained from the modified speed map computed in the FM-1st step. The resulting arrival time map is shown in Fig. 2D.

4. Finally, a gradient descent is applied over the resulting arrival time map from the starting point to the goal point. An optimal path in terms of the arrival time, smoothness, and safety is obtained.

## Gradient descent method

The global path can be extracted by applying the gradient descent method. The path propagates along the gradient descent direction from the starting position $P_s$ to the goal position $P_g$ with a step length $d$. The value of $d$ can be set to a value equal to the SR of the

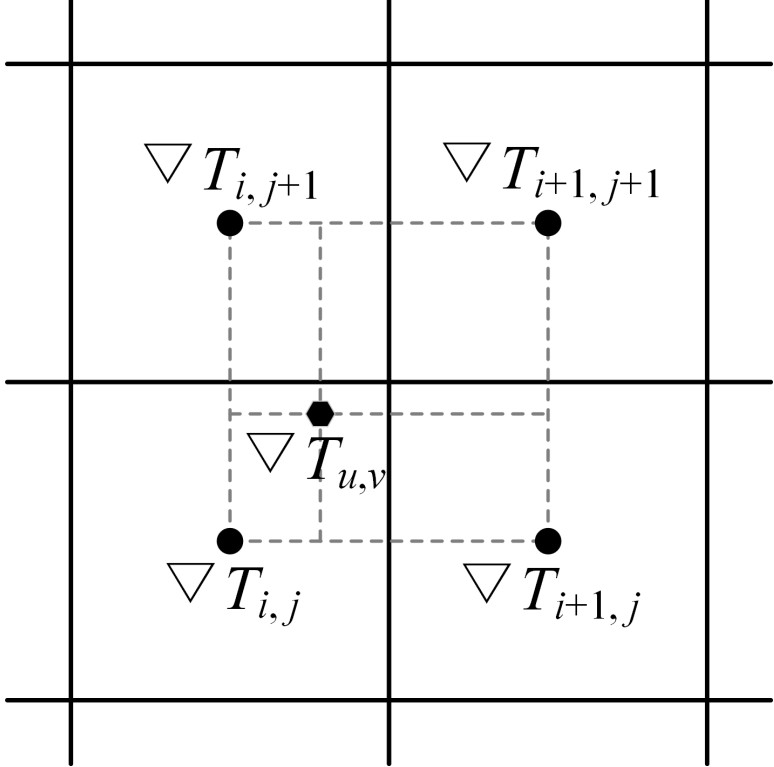

**Figure 3  Modified gradient of a path waypoint calculated by the bilinear interpolation method.**

grid map. The gradient of the cell $x_{i,j}$ is calculated by:

$$\nabla T_{i,j} = \left[ \frac{T_{i+1,j} - T_{i-1,j}}{2} \quad \frac{T_{i,j+1} - T_{i,j-1}}{2} \right]^{\mathrm{T}} \tag{5}$$

where $T_{i,j}$ represents the arrival time value at cell $x_{i,j}$ and the non-italic T in the upper-right corner represents the transpose of vector $[x\ y]$.

The path waypoints are not limited to cell centers. Generally, the gradient of a path waypoint $P_n = (u, v)$ located in the cell $x_{i,j}$ is approximated by the cell gradient $\nabla T_{i,j}$. However, a modified gradient $\nabla T_{u,v}$ calculated by the bilinear interpolation can be used to improve the path precision (see Fig. 3).

## PROPOSED ALGORITHM

To improve the computational efficiency of global path planning in a large-scale and multi-island marine environment, the proposed algorithm performs path planning twice for different purposes. First, the path planning is performed in a low SR (LSR) grid map to determine an effective region. The final global path is then obtained in the second path planning within the effective region of a high SR (HSR) grid map. The relevant methods and procedures applied in the algorithm are as follows. The complete algorithm flow is summarized in the last part of this section.

## Mapping of two-level SR grid maps

The two-level SR grid maps are contained in the HSR and LSR grid maps. The HSR grid map is directly obtained from the Google satellite image data. A mapping relationship between the LSR cells and the corresponding $L \times L$ HSR cell sub-blocks is established, which can be expressed as:

$$(i_\mathrm{L}, j_\mathrm{L}) \sim \begin{bmatrix} (i'_\mathrm{H}, j'_\mathrm{H}+L-1) & \cdots & (i'_\mathrm{H}+L-1, j'_\mathrm{H}+L-1) \\ \vdots & \ddots & \vdots \\ (i'_\mathrm{H}, j'_\mathrm{H}) & \cdots & (i'_\mathrm{H}+L-1, j'_\mathrm{H}) \end{bmatrix} \tag{6}$$

where $(i_\mathrm{L}, j_\mathrm{L})$ are the LSR cell coordinates, and $(i'_\mathrm{H}, j'_\mathrm{H})$ are the original cell coordinates of the mapped HSR cell sub-block, as shown in Fig. 4.

To map the LSR grid map to a sub-block of the HSR grid map, the following is used:

$$\begin{cases} i'_\mathrm{H} = i_\mathrm{Ho} + L i_\mathrm{L} \\ j'_\mathrm{H} = j_\mathrm{Ho} + L j_\mathrm{L} \end{cases} \tag{7}$$

To map from a sub-block of the HSR grid map to a cell of the LSR grid map, we use:

$$\begin{cases} i_\mathrm{L} = f_\mathrm{floor}\left(\dfrac{i_\mathrm{H} - i_\mathrm{Ho}}{L}\right) \\ j_\mathrm{L} = f_\mathrm{floor}\left(\dfrac{j_\mathrm{H} - j_\mathrm{Ho}}{L}\right) \end{cases} \tag{8}$$

where $(i_\mathrm{H}, j_\mathrm{H})$, $i'_\mathrm{H} \le i_\mathrm{H} \le i'_\mathrm{H}+L-1$, $j'_\mathrm{H} \le j_\mathrm{H} \le j'_\mathrm{H}+L-1$ are the HSR cell coordinates. $(i_\mathrm{Ho}, j_\mathrm{Ho})$ are the original cell coordinates of the original HSR cell sub-block (see Fig. 4), which are determined by:

$$\begin{cases} i_\mathrm{Ho} = \left(i_\mathrm{Hg} - f_\mathrm{floor}\left(\dfrac{L}{2}\right)\right) \% L \\ j_\mathrm{Ho} = \left(j_\mathrm{Hg} - f_\mathrm{floor}\left(\dfrac{L}{2}\right)\right) \% L \end{cases} \tag{9}$$

where $(i_\mathrm{Hg}, j_\mathrm{Hg})$ are the HSR cell coordinates of the goal cell $x_\mathrm{g}$, the function $f_\mathrm{floor}(m)$ indicates a rounding down of $m$, and $a\%b$ indicates the modulo operation.

The SR of the LSR grid map is:

$$D_\mathrm{LRes} = L D_\mathrm{HRes} \tag{10}$$

where $D_\mathrm{HRes}$ is the SR of the HSR grid map. In general, $L$ is defined as:

$$L \le f_\mathrm{round}\left(\frac{D_\mathrm{Th}}{2 D_\mathrm{HRes}}\right) \tag{11}$$

where $D_\mathrm{Th}$ is the distance threshold of the virtual obstacle influence, and the function $f_\mathrm{round}(m)$ indicates a rounding of $m$. The cell numbers of the LSR grid map in the $X$-axis and $Y$-axis directions are:

$$M_\mathrm{L} = f_\mathrm{floor}\left(\frac{M_\mathrm{H} - i_\mathrm{Ho}}{L}\right) \tag{12}$$

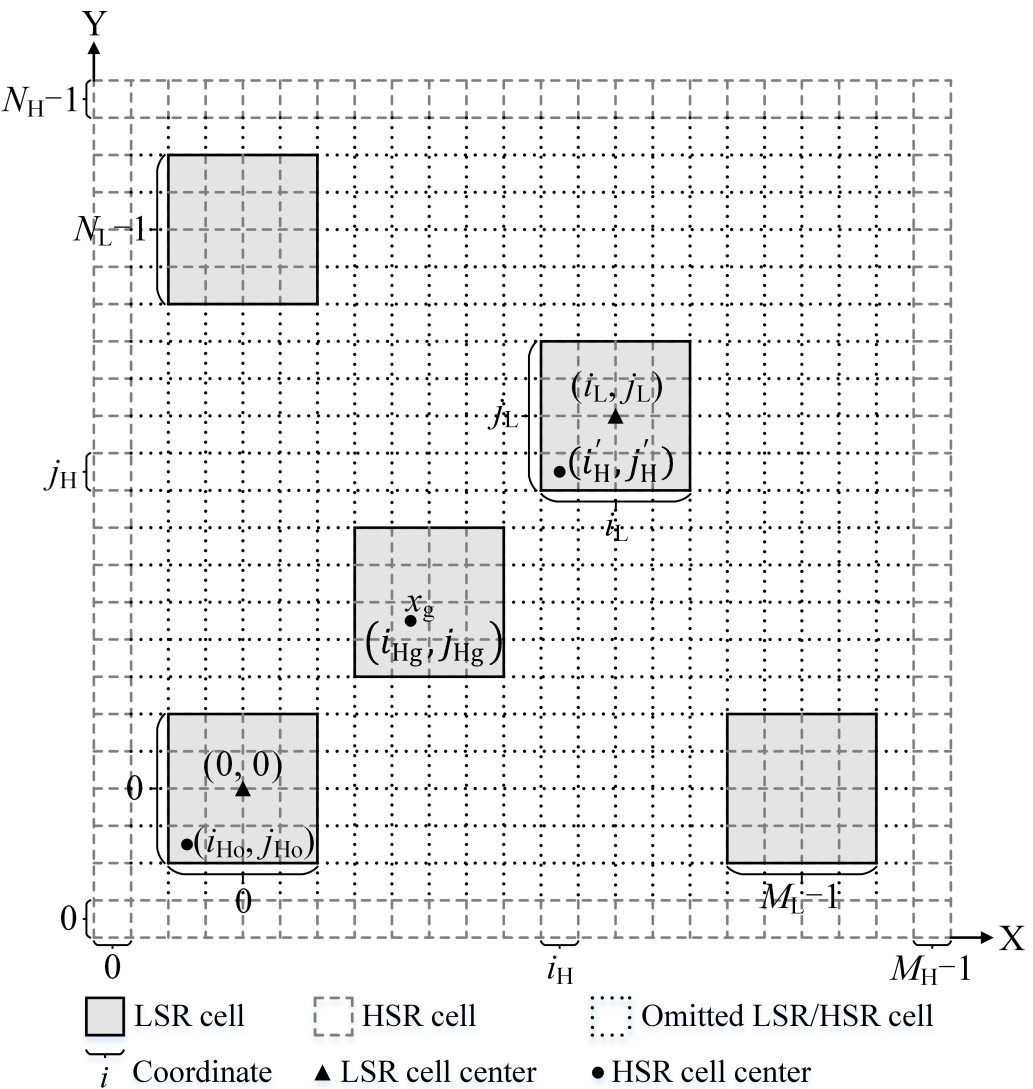

**Figure 4  Mapping between an LSR grid map and an HSR grid map.** Here, a case with $L = 4$ is shown. $(i_{Ho}, j_{Ho})$ is adaptively fine-tuned based on $x_g$. Therefore, $x_g$ is set as the center ($L$ is odd) or the quasi-center ($L$ is even) cell of an HSR cell sub-block.

$$N_L = f_{floor}\left(\frac{N_H - j_{Ho}}{L}\right) \tag{13}$$

where $M_H$ and $N_H$ are the cell numbers of the HSR grid map in the $X$-axis and $Y$-axis directions, respectively.

Based on the mapping relationship, an LSR cell type is determined by comparing the setting threshold $\Gamma$ and obstacle cell proportion $\gamma$ in the HSR cell sub-block. If $\gamma > \Gamma$, the LSR cell is set as an obstacle cell; otherwise, it is set as a free cell. $\Gamma$ prefers to select a small value to retain as much obstacle information as possible.

## IDC-FM² method

In the basic FM² method, the path is improved using a modified speed map. Two parameters, $\alpha$ and $\beta$, are used to modify the speed map. These modifications ensure the flexibility of the computed path. However, these two parameters have no clear physical meaning. The normalization of the speed values in the FM-1st step also results in several shortcomings. For example, all the speed values in the speed map must be calculated. When the map range changes slightly, the entire speed map rescales and differs in the same locations of the original map, and the parameters must be adjusted.

An IDC-FM² method is proposed to address these shortcomings. In this method, a time-cost weighting function, $w_x$, is introduced to adjust the speed map. Three inshore-distance parameters (the distance threshold $D_{Th}$, as well as the virtual strong and weak constraint distance parameters, $D_{sc}$ and $D_{wc}$, respectively) and two weighting function values with respect to $D_{sc}$ and $D_{wc}$ ($w_{sc}$ and $w_{wc}$, respectively) are used to determine the function $w_x$, and $D_{Th}$ is used to achieve the same function as the speed saturation. Although five parameters are set, only two inshore-distance parameters, $D_{Th}$ and $D_{sc}$, must be considered, while suitable values of $w_{sc}$ and $w_{wc}$ can be selected as constant values. Based on this method, the path portion around an obstacle is constrained in a region determined by $D_{Th}$ and $D_{sc}$.

### *Time-cost weighting function*

The time-cost weighting function $w_x$ is introduced as a concept of the virtual obstacle influence. When a USV is far away from obstacles, there is no obstacle influence. As the USV approaches, the virtual obstacle influence increases slowly at first and the amplitude increases gradually. The obstacle influence increases dramatically when the USV is close to within a certain degree.

Similar to the FM-1st step in the basic FM² method, all obstacle cells are set as source cells to first calculate the arrival time map. The difference here is that a threshold $T_{Th} = \tau D_{Th}/D_{Res}$ could be set to speed up the calculation, where $D_{Res}$ is the SR of the map, and $\tau$ is the unified time-cost which can take 1 as the value. In the loop procedure, if the smallest $T$ value of a cell in $\mathcal{S}_T$ is larger than or equal to $T_{Th}$, all $T$ values of non-accepted cells can be set as $T_{Th}$ directly. Based on the resulting arrival time map, $\mathcal{T}_{sat}$, with a saturation threshold, the time-cost weighting function map, $\boldsymbol{W}'$, is designed as:

$$w'_x = \begin{cases} \infty, & T_x = 0 \\ 1 + a\left(\dfrac{T_{Th}}{T_x} - 1\right)^b, & T_x > 0 \end{cases} \tag{14}$$

where $w'_x$ and $T_x$ are the values of the cell $x$ in $\boldsymbol{W}'$ and $\mathcal{T}_{sat}$, respectively, and $a$ and $b$ are two positive coefficients to be determined.

When a free cell, $x$, is within the virtual influenced scope of obstacles (i.e., $0 < D_x \leq D_{Th}$, where $D_x$ is the closest distance to obstacles), then:

$$D_x \approx D_{Res}\frac{T_x}{\tau} \tag{15}$$

Therefore, $w_x$ with respect to $D_x$ is used to approximate $w'_x$:

$$w_x(D_x) = \begin{cases} \infty, & D_x = 0 \\ 1 + a\left(\dfrac{D_{\mathrm{Th}}}{D_x} - 1\right)^b, & 0 < D_x \le D_{\mathrm{Th}} \\ 1, & D_x > D_{\mathrm{Th}} \end{cases} \tag{16}$$

where $w_x(D_x)$ is the value in cell $x$ of the approximate time-cost weighting function map, $W$. The time-cost value of each cell in the time-cost map, which is the inversion of the speed map, is then adjusted to:

$$new\,\tau_x = w_x(D_x)\,\tau_x \tag{17}$$

Equation 16 implies that $w_x$ increases when cell $x$ is closer to obstacles. The planned path tends to select points with a lower arrival time-cost. Therefore, when the path is close to obstacles, it will be farther away from obstacles compared to the path planned by the basic FMM.

### Parameters for the time-cost weighting function

Five parameters, $D_{\mathrm{Th}}$, $D_{\mathrm{sc}}$, $D_{\mathrm{wc}}$, $w_{\mathrm{sc}}$, and $w_{\mathrm{wc}}$, are introduced to determine the coefficients $a$ and $b$, which are determined by:

$$b = \frac{\ln(w_{\mathrm{sc}} - 1) - \ln(w_{\mathrm{wc}} - 1)}{\ln(1 - e_{\mathrm{sc}}) - \ln(1 - e_{\mathrm{wc}}) + \ln e_{\mathrm{wc}} - \ln e_{\mathrm{sc}}} \tag{18}$$

$$a = (w_{\mathrm{sc}} - 1)\left(\frac{e_{\mathrm{sc}}}{1 - e_{\mathrm{sc}}}\right)^b \tag{19}$$

where $e_{\mathrm{sc}} = D_{\mathrm{sc}}/D_{\mathrm{Th}}$, $e_{\mathrm{wc}} = D_{\mathrm{wc}}/D_{\mathrm{Th}}$, $D_{\mathrm{sc}} < D_{\mathrm{wc}} < D_{\mathrm{Th}}$, and $w_{\mathrm{sc}} > w_{\mathrm{wc}} > 1$.

The parameter $D_{\mathrm{Th}}$ determines the largest virtual influence scope of the obstacles. The suggestion for the selection of $D_{\mathrm{Th}}$ is that there should be a sufficiently safe buffer area for obstacles. The parameter $D_{\mathrm{sc}}$ is very important for the safety of the USV. First, it is suggested that this parameter satisfies:

$$D_{\mathrm{sc}} \ge v_{\mathrm{U,max}} t_{\mathrm{r}} - \frac{v_{\mathrm{U,max}}^2}{2a_{\mathrm{d}}} \tag{20}$$

where $v_{\mathrm{U,max}}$ is the maximum speed of the USV, $t_{\mathrm{r}}$ is the reaction time of the thruster, and $a_{\mathrm{d}}$ is the negative maximum acceleration of the USV under braking. In an unknown environment, it is better to select a slightly larger value to ensure safety, which is similar in environments with shallow and reef waters. This value may be relatively small in deep and reef-free waters to improve path efficiency. $D_{\mathrm{wc}}$ is a parameter that acts as the adjusted constraint distance. In a non-channel ocean area, the path portions around obstacles will be limited to the regions between $D_{\mathrm{wc}}$ and $D_{\mathrm{Th}}$. The $D_{\mathrm{wc}}$ value can be set independently or determined jointly by $D_{\mathrm{Th}}$ and $D_{\mathrm{sc}}$. In this study, it is set as:

$$D_{\mathrm{wc}} = D_{\mathrm{Th}} - \frac{\sqrt{2}}{2}(D_{\mathrm{Th}} - D_{\mathrm{sc}}) \tag{21}$$

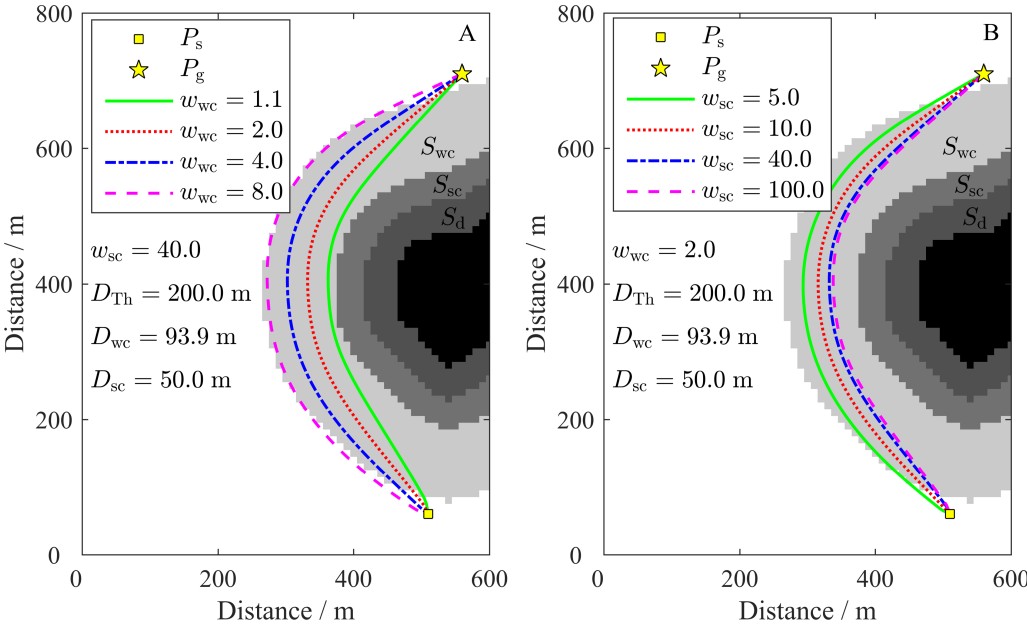

**Figure 5   Influence of $w_{wc}$ and $w_{sc}$ on the path.** (A) Influence of $w_{wc}$ on the path. The black region is the obstacle, while the regions with different grayscales from dark to light are the danger region, $S_d$; the strong constraint region, $S_{sc}$; and the weak constraint region, $S_{wc}$. (B) Influence of $w_{sc}$ on the path.

The area around the obstacles is divided into four parts by the three inshore-distance parameters: $D_{sc}$, $D_{wc}$, and $D_{Th}$. The three parts extending from an obstacle to the outside region are the danger region, $S_d$; the strong constraint region, $S_{sc}$; and the weak constraint region, $S_{wc}$ (see Fig. 5). The influences of $w_{wc}$ and $w_{sc}$ on the path are shown in Fig. 5. As shown in Fig. 5A, the path will be located far from the obstacle until it is near the boundary of $S_{wc}$ when $w_{wc}$ is large. In contrast, the path is closer to the obstacle when $w_{sc}$ is large (see Fig. 5B). The minimum degree of the path close to $S_{sc}$ is mainly determined by $w_{wc}$, which can be inferred by further comparing the paths in Fig. 5B with the closest path to the obstacle ($w_{wc} = 1.1$) in Fig. 5A. However, regardless of the change in $w_{wc}$ and $w_{sc}$ ($w_{sc} > w_{wc} > 1$), the path portions bypassing obstacles will always be within $S_{wc}$ or around the outside boundary of $S_{wc}$ in non-channel ocean areas. In channel areas where there is no $S_{wc}$, the path will follow the quasi-centerline of the channel if it passes through the channel. In this study, the values of $w_{sc} = 40.0$ and $w_{wc} = 2.0$ were selected and fixed. The paths can be flexibly modified by the inshore-distance parameters $D_{Th}$ and $D_{sc}$.

## Determination of effective regions within the HSR grid map

The effective region within the HSR grid map is important for improving the computational efficiency of the proposed algorithm. It is acquired based on the mapping between the LSR and HSR grid maps when the corresponding region within the LSR grid map is determined.

Two effective regions that are respectively used in the FM-1st and FM-2nd steps of the IDC-FM$^2$ method must be determined. The low-precision initial path, $\ell_{ini}$, is obtained first based on the LSR arrival time map $\mathcal{T}_L$. Two effective regions within the LSR grid map,

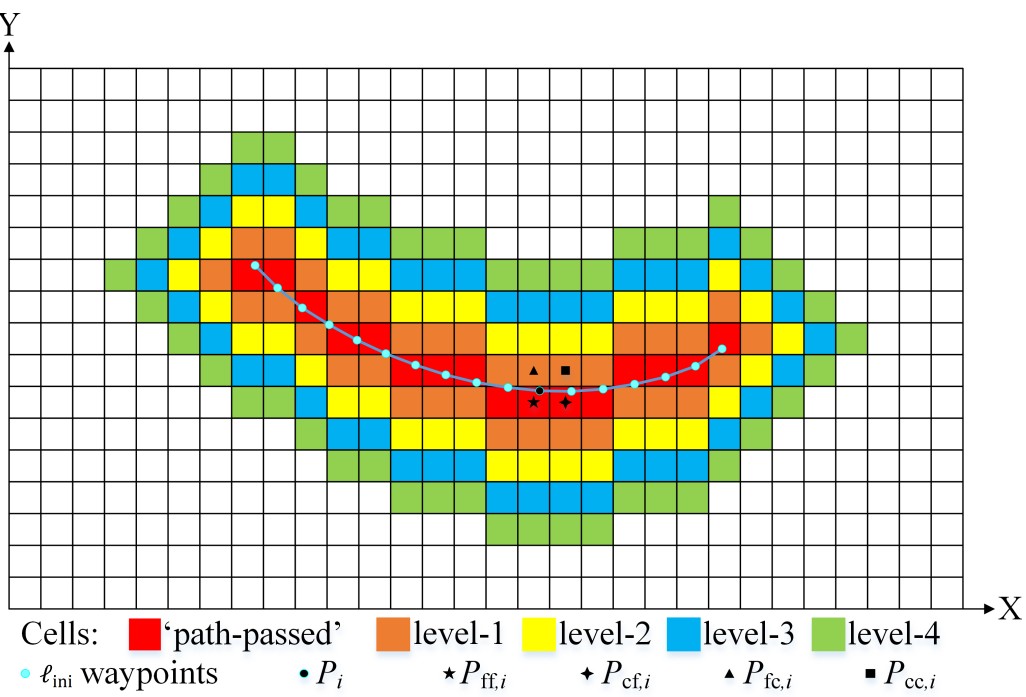

**Cells:** ■ 'path-passed'  ■ level-1  ■ level-2  ■ level-3  ■ level-4
○ $\ell_{\text{ini}}$ waypoints    • $P_i$    ★ $P_{\text{ff},i}$    ✦ $P_{\text{cf},i}$    ▲ $P_{\text{fc},i}$    ■ $P_{\text{cc},i}$

**Figure 6** **Effective region within the LSR grid map.** The case shown uses $k = 4$, and $P_{ff,i} = (f_{\text{floor}}(x_i),$ $f_{\text{floor}}(y_i))$, $P_{cf,i} = (f_{\text{ceil}}(x_i), f_{\text{floor}}(y_i))$, $P_{cc,i} = (f_{\text{ceil}}(x_i), f_{\text{ceil}}(y_i))$, and $P_{fc,i} = (f_{\text{floor}}(x_i), f_{ceil}(y_i))$ are the four neighbor cells of $P_i$. $P_{ff,i}$ is the "path-passed" cell with respect to $P_i$ in this case.

$S_{\text{LER\_1st}}$ and $S_{\text{LER\_2nd}}$, are determined by expanding the cells around the "path-passed" cells of $\ell_{\text{ini}}$. As shown in Fig. 6, the nearest neighbor cell among the four neighbor cells of the waypoint $P_i = (x_i, y_i)$ is defined as the "path-passed" cells. When all "path-passed" cells are determined, the $\kappa_{\text{2nd}} = \kappa$ levels of the cells are extended to obtain $S_{\text{LER\_2nd}}$. Considering $S_{\text{LER\_1st}}$, there are three different situations:

- **Situation** 1: there is no cell with $w_x > 1$ in $S_{\text{LER\_2nd}}$;
- **Situation** 2: there is at least one cell with $w_x > 1$ and no obstacle cell in $S_{\text{LER\_2nd}}$.
- **Situation** 3: there is at least one obstacle cell in $S_{\text{LER\_2nd}}$.

In **Situations** 1 and 3, $\kappa_{1st} = \kappa$. In **Situation** 2, $S_{\text{LER\_2nd}}$ is extended continuously with $\Delta\kappa$ levels of cells until there is at least one obstacle cell for the first time, and $\kappa_{1st} = \kappa + \Delta\kappa$. Finally, two corresponding effective regions within the HSR grid map, $S_{\text{HER\_1st}}$ and $S_{\text{HER\_2nd}}$, are determined and used in the different FM steps of the IDC-FM$^2$ method to obtain the HSR arrival time map $\mathcal{T}_{\text{H}}$.

The extended parameter $\kappa$ affects the paths (recorded as $\ell_\kappa$) based on the proposed algorithm. When $\kappa$ is not adequately large, these paths may not be consistent with the reference path $\ell_{\text{ref}}$, which is acquired by applying the IDC-FM$^2$ method in the HSR grid map directly. A navigable global path is used as a case to compare the consistency between $\ell_\kappa$ and $\ell_{\text{ref}}$. The basic parameters used in the proposed algorithm are listed in Table 1. Using $\ell_{\text{ref}}$ as the reference, the distances between the corresponding waypoints of $\ell_\kappa$ ($\kappa = 3, \ldots, 6$)

**Table 1   Basic parameter values for the proposed algorithm.**

| Parameter | Value | Parameter | Value | Parameter | Value |
|---|---|---|---|---|---|
| $L$ | 8 | $\Gamma$ | 0.2 | $D_{\text{Th}}$ (m) | 200 |
| $D_{\text{sc}}$ (m) | 50 | $w_{\text{sc}}$ | 40.0 | $w_{\text{wc}}$ | 2.0 |

and $\ell_{\text{ref}}$ are shown in Fig. 7. It is shown that the largest distance decreases when $\kappa$ increases, and the distance becomes 0 when $\kappa \geq 7$. Other cases were tested and similar results were obtained. These results indicate that there is good consistency when $\kappa$ is sufficiently large (such as $\kappa = 10$). $\Delta\kappa$ is a variable parameter that was determined based on three different cases.

## Algorithm flow

The proposed algorithm is improved based on the basic $FM^2$-based algorithm. The basic $FM^2$-based algorithm is executed directly on a single grid map. The algorithm flow of this basic algorithm is shown in Fig. 8 in the form of main data stream and used methods. This algorithm flow has two main steps. In *Step* S1, an arrival time map with obstacle sources and a saturation threshold, $\mathcal{T}_{\text{sat}}$, is obtained by first applying the FMM. Then, the approximate time-cost weighting function map, $W$, is calculated and used to adjust the time-cost map. Based on the adjusted time-cost map, an arrival time map with the goal point source, $\mathcal{T}$, is obtained by applying the FMM again. The entire process is used to obtain $\mathcal{T}$ by applying the IDC-$FM^2$ method. Finally, the global path is acquired based on $\mathcal{T}$ by applying the gradient descent method in *Step* S2.

The main data stream in the proposed algorithm and the methods used to obtain them are shown in Fig. 9. The main flow is as follows:

1. *Step* T1. Obtain the two-level SR grid maps;
2. *Step* T2. Obtain the LSR arrival time map $\mathcal{T}_{\text{L}}$ in the LSR grid map;
3. *Step* T3. Determine the effective regions within the HSR grid map ($S_{\text{HER\_1st}}$ and $S_{\text{HER\_2nd}}$);
4. *Step* T4. Obtain the HSR grid arrival time map $\mathcal{T}_{\text{H}}$ with the effective region constraint in both FM steps of the IDC-$FM^2$ method. There are several adjustments in both FM steps compared to the basic FMM performance. For the FM-1st step, all $w_x$ values within $S_{\text{HER\_1st}}$ need not be adjusted, and this step can be ignored in *Situation* 1. In *Situation* 2 and *Situation* 3, all free cells out of $S_{\text{HER\_1st}}$ are set to $\mathcal{S}_{\text{A}}$ with $T = \infty$ in the initialization of this step. In the FM-2nd step, all free cells out of $S_{\text{HER\_2nd}}$ are set to $\mathcal{S}_{\text{A}}$ with $T = \infty$ in the initialization;
5. *Step* T5. Acquire the final high precision global path by applying the gradient descent method based on $\mathcal{T}_{\text{H}}$.

## RESULTS AND DISCUSSION

### Simulation environments

Two surrounding spatial areas of Zhucha Island and Changhai County were selected as the simulation environments. The original maps adopted Google satellite images, as shown in

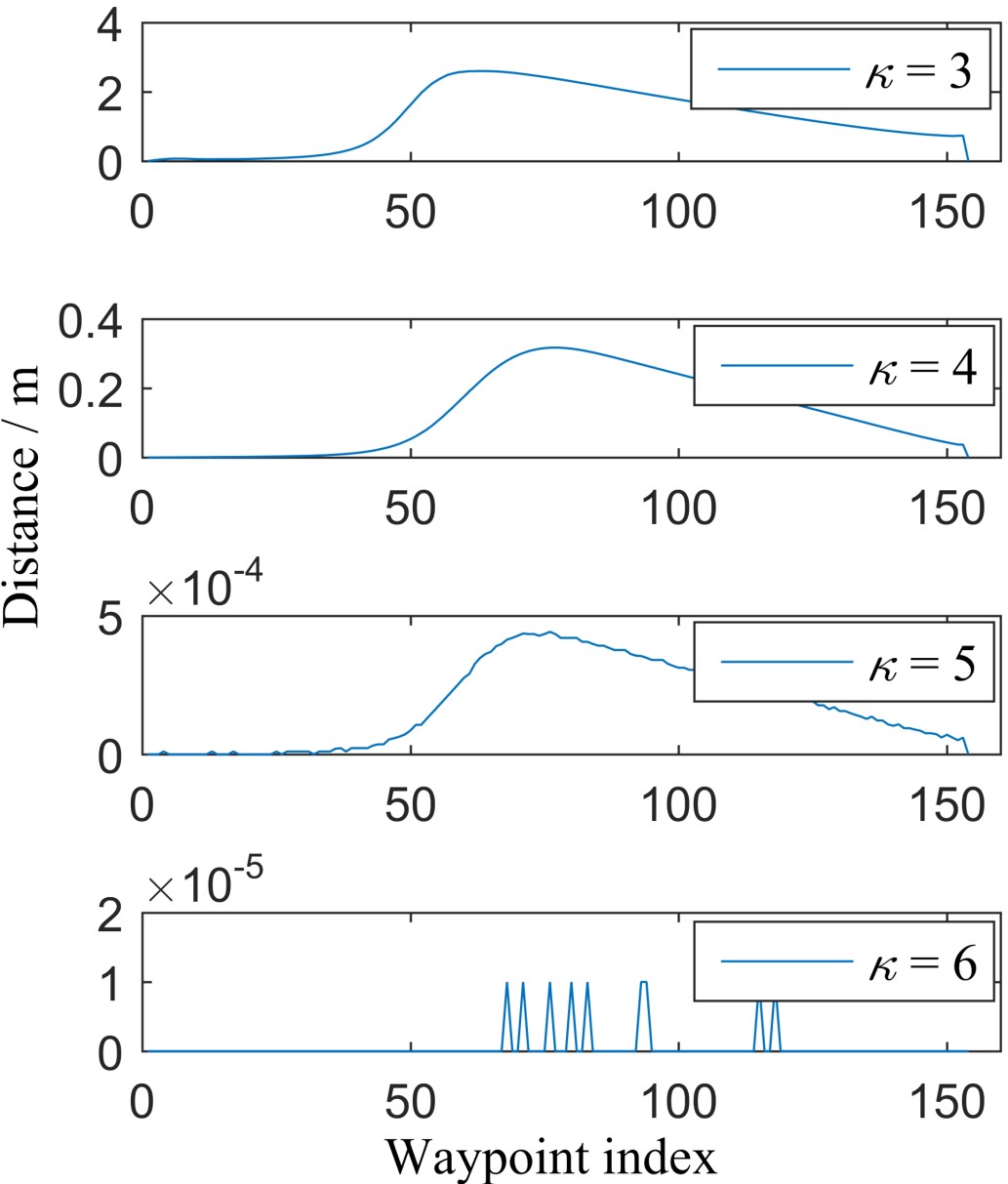

**Figure 7** **Distances between the corresponding waypoints of $l_{ref}$ and $l_k$ ($k = 3, \ldots, 6$).** $l_{ref}$ is the path obtained by applying the IDC-FM$^2$ method based on the single HSR grid map directly, while $l_k$ is the path obtained by applying the proposed algorithm based on two-level SR grid maps.

Fig. 10. The spatial resolutions in the longitude and latitude directions are approximately 4.8 m and 3.9 m, respectively. Temporary binary grid maps are obtained based on the image processing method (*Shi et al., 2018*) combined with manual assistance, and then the HSR grid maps with $D_{HRes} = 10m$ in both the longitude and latitude directions are determined by resampling processing. The corresponding binary grid maps are presented in Fig. 11. Their ranges are 7 km × 7 km and 64 km × 48km, respectively.

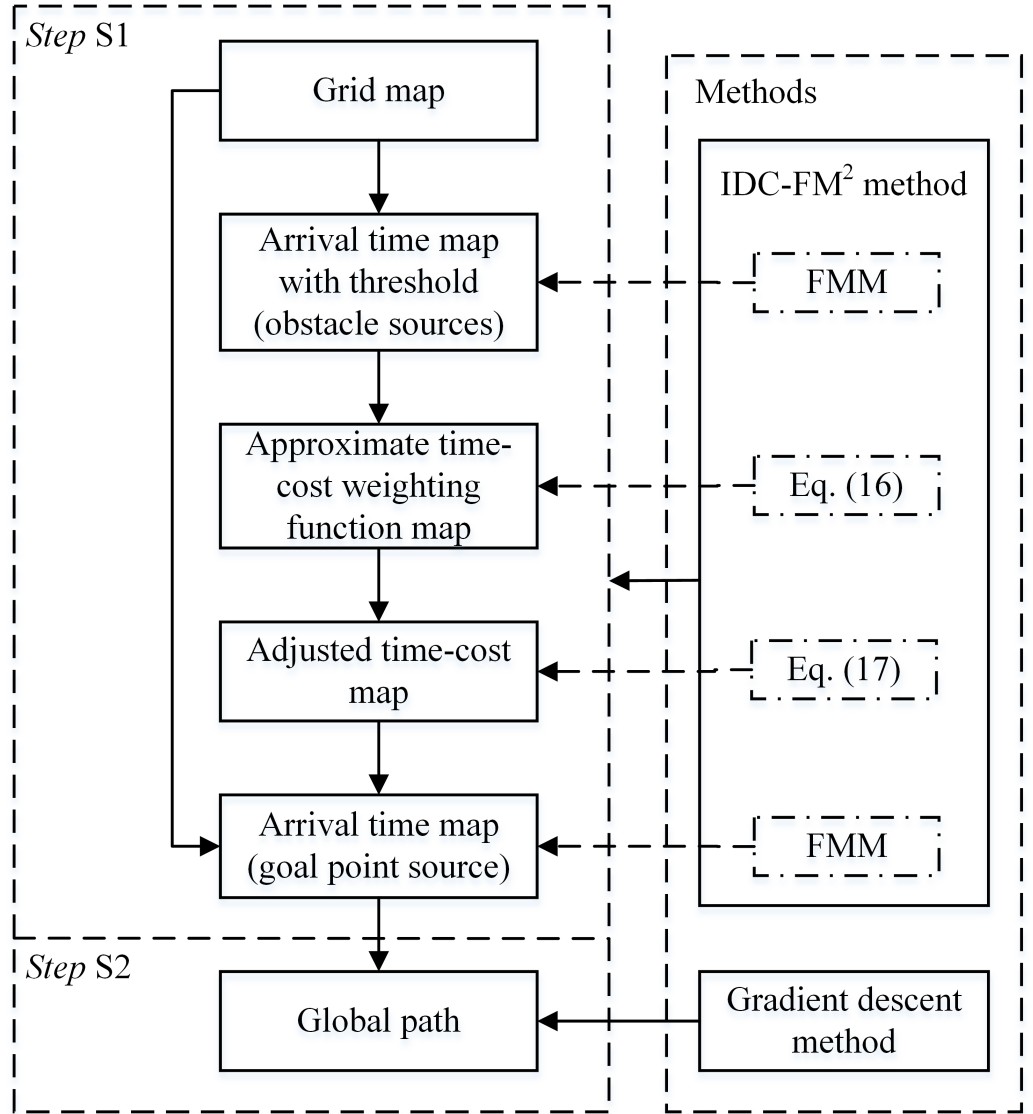

**Figure 8  Algorithm flow of the basic algorithm applying the IDC-FM² method on a single grid map.**
This flow is expressed in the form of main data stream and used methods.

## Inshore-distance-constraint performances

A path planning from $P_s = [4.3$ km, $2.8$ km$]$ to $P_g = [3.6$ km, $2.0$ km$]$ is used as a typical case to analyze the inshore-distance-constraint performance of the IDC-FM² method using the inshore-distance parameters $D_{Th}$ and $D_{sc}$. As shown in Fig. 12, the path planned based on the basic FMM, $\ell_{FMM}$, is very close to the islands when this path bypasses them. For comparison, all paths planned by the proposed algorithm, $\ell_1$ to $\ell_4$, are located away from islands by a certain distance. They are therefore significantly better choices from a safety perspective.

Paths $\ell_1$ to $\ell_4$ are acquired using different inshore-constraint distance parameters ($D_{Th}$ and $D_{sc}$, see Table 2 wherein $D_{wc}$ is calculated based on $D_{Th}$ and $D_{sc}$) in the IDC-FM²

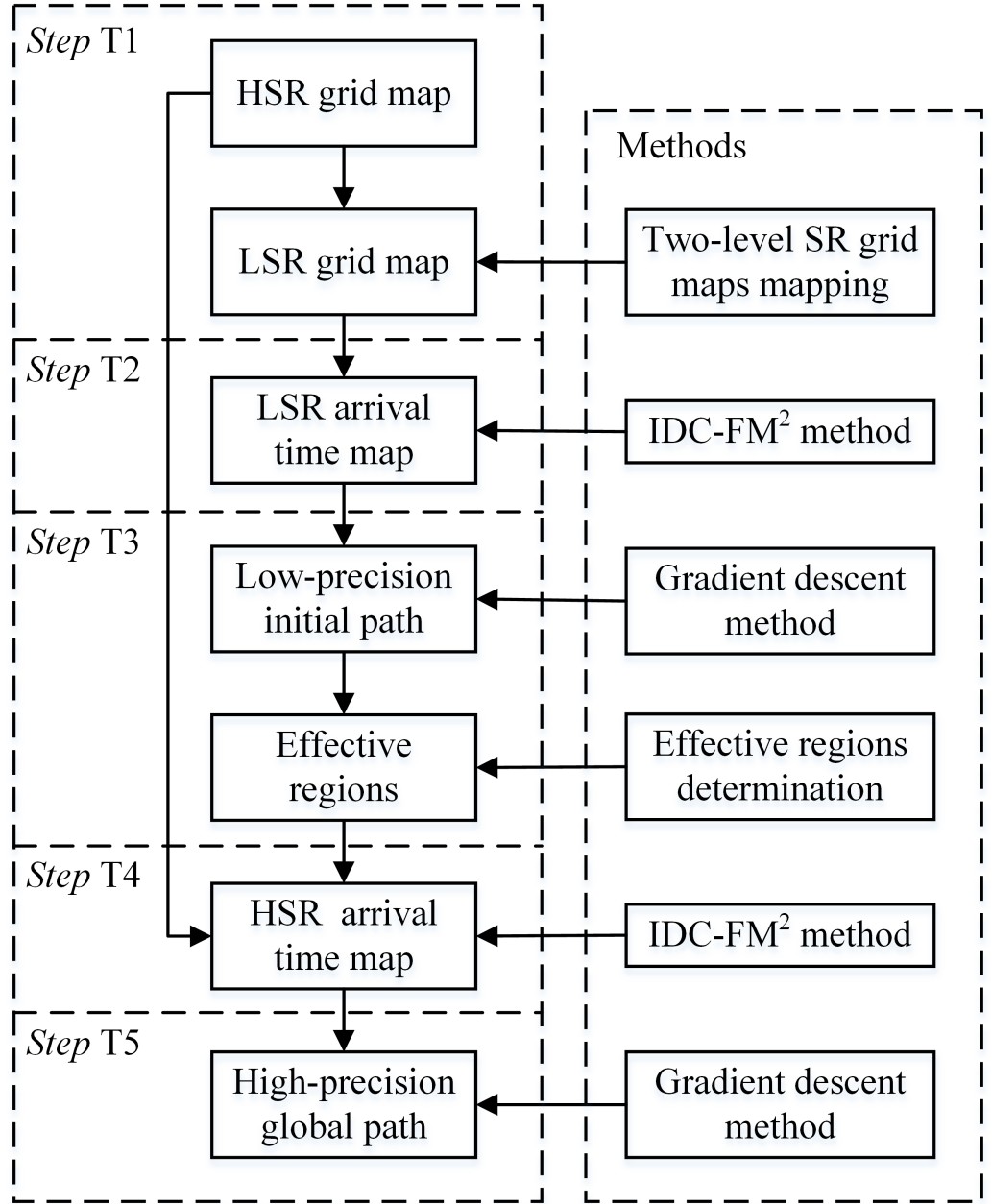

**Figure 9** **Algorithm flow of the proposed algorithm performed on two-level SR grid maps.** This flow is expressed in the form of main data stream and used methods.

method in which $w_{sc} = 40.0$ and $w_{wc} = 2.0$. These paths are clearly adjusted by these distance parameters. When the distance constraints ($D_{Th}$ and $D_{sc}$) are small, the path (such as $\ell_1$, see Fig. 12) will be a somewhat close to the islands. The estimated quasi-closest distance from $\ell_1$ to an island is approximately 38 m, which is shorter than half of the shortest channel width ($d_{hscw} \approx 101m$). In this situation, $\ell_1$ is always outside of its $S_{sc}$ (<28.2 m) value, and the path portions around the islands are located in the region of

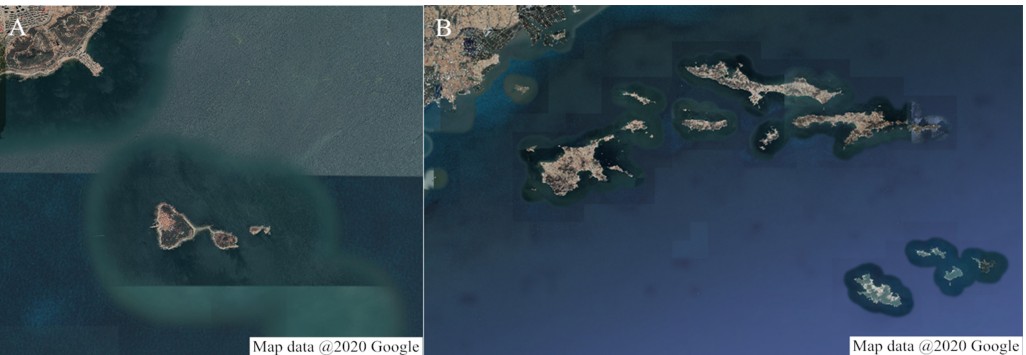

**Figure 10  Google satellite images of the simulation environments.** (A) Surrounding areas of Zhucha Island in Qingdao, Shandong Province, China. Map data @ 2020 Google. (B) Surrounding areas of Changhai County in Liaoning Province, China. Map data @ 2020 Google.

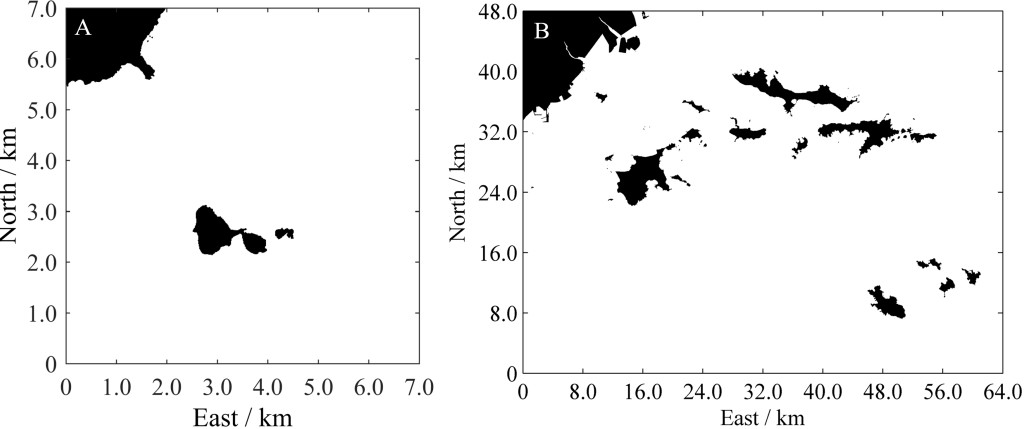

**Figure 11  Binary grid maps of the simulation environments.** (A) Binary grid map of the environment shown in Fig. 10A. The black cells indicate obstacle areas, while the white cells indicate the navigable areas for USVs. (B) Binary grid map of the environment shown in Fig. 10B.

its $S_{wc}$ (28.2 m to 60.0 m). When the distance constraints increase, the paths (such as $\ell_2$ and $\ell_3$, see Fig. 12) in the non-channel areas will be outside of $S_{sc}$. The estimated quasi-closest distances (approximately 115 m and 128 m, respectively) are larger than $d_{hscw}$. However, $\ell_2$ and $\ell_3$ will be along the quasi-midline of the channel in the channel area, regardless of whether $d_{hscw}$ is greater or less than $D_{wc}$ for these two paths because the weighting time-cost will be smaller than the path around the islands (as in $\ell_4$, see Fig. 12). If the distance constraints are further increased, $\ell_4$ will be selected for safety, although more time will be required. These results indicate that the path based on the proposed algorithm can be adjusted by setting different inshore-constraint distance parameters to meet safety requirements. Additionally, when the inshore-constraint distance parameters are determined, the path will cross a channel when its width is adequately large; otherwise, the path will bypass around the islands.

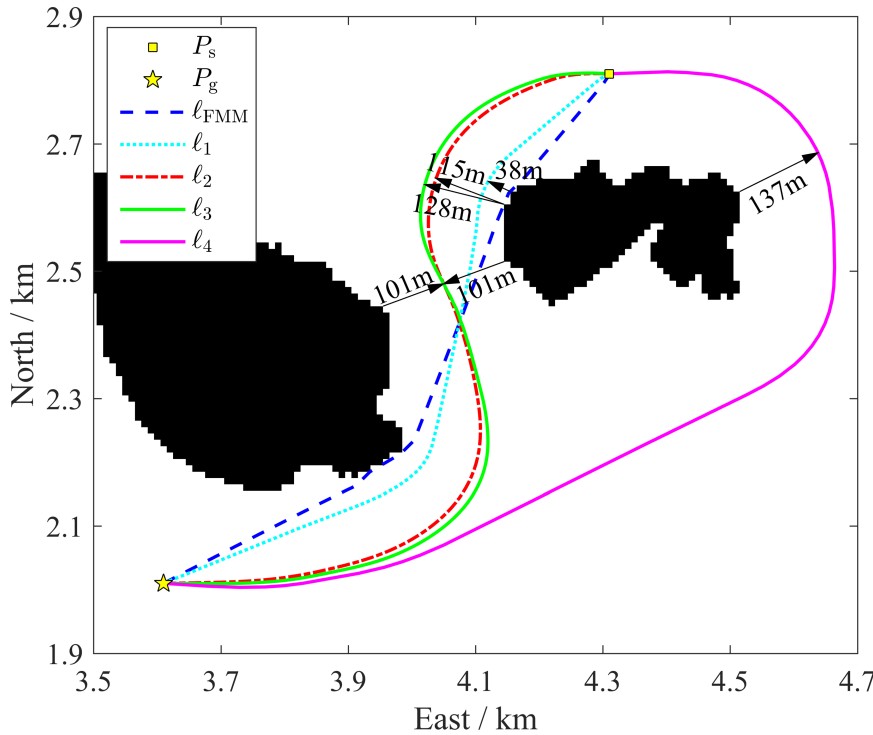

**Figure 12 Path based on the basic FMM ($l_{\text{FMM}}$) and paths based on the proposed algorithm with different parameter settings of $w_x$ ($l_1$ to $l_4$).** The indicated distances are the estimated quasi-closest distances between the islands and path segments.

**Table 2 Inshore-constraint distance parameters for the determination of $w_x$.**

| Path | $D_{\text{Th}}$ (m) | $D_{\text{wc}}$ (m) | $D_{\text{sc}}$ (m) |
|---|---|---|---|
| $\ell_1$ | 60.0 | 28.2 | 15.0 |
| $\ell_2$ | 200.0 | 79.8 | 30.0 |
| $\ell_3$ | 200.0 | 104.5 | 65.0 |
| $\ell_4$ | 200.0 | 118.7 | 85.0 |

One classical approach to plan collision-free paths is applying the FMM within the map with inflated obstacles. For comparison, paths planned by the classical approach and the IDC-FM² method are shown in Fig. 13. The starting and goal positions are $P_s = [4.15$ km, $2.74$ km] to $P_g = [4.33$ km, $2.40$ km] and the island obstacles have been inflated by 94.0 m. As shown in Fig. 13, the path planned by the classical approach, $\ell_{\text{FMM+inflated}}$, is obviously influenced by the shape of the inflated obstacle. In some path turns which are marked by ellipses in Fig. 13, they are both affected by the shape of the inflated obstacle and the time-value gradient calculation of cells adjacent to the inflated obstacle. A smooth path, $\ell_{\text{FMM+inflated+smooth}}$, is also shown in Fig. 13. Although it removes abrupt turn waypoints, the path is still not very smooth. As comparisons, paths planed based on the IDC-FM² method, $\ell_1$ and $\ell_2$, are obviously smoother than $\ell_{\text{FMM+inflated+smooth}}$. In addition, when using the proposed rapid path planning algorithm based on two-level SR grid maps to

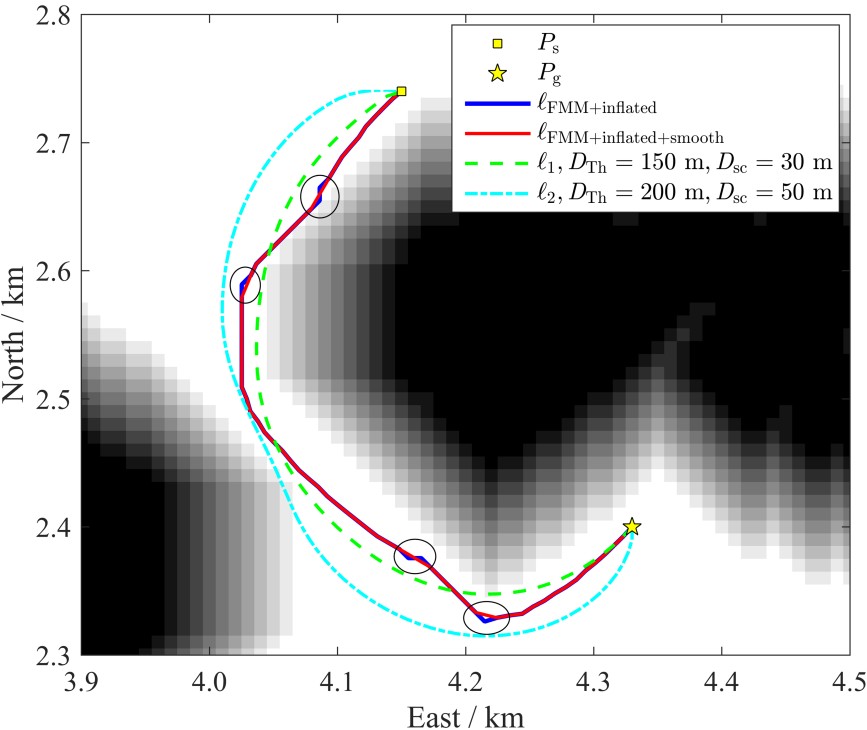

**Figure 13** **Paths planned by a classical approach applying the FMM within the map with inflated obstacles and the IDC-FM² method.** When using the classical approach, the obstacle is inflated by 94.0 m. The black cells indicate the raw obstacle, the white cells indicate the free areas, while the cells with different gray scales indicate the inflated obstacles.

improve the computational efficiency, the channel will be mapped as obstacle cells when mapping the HSR grid map to the LSR grid map in this case (taking the mapping parameter $L = 8$ as shown in Table 1), because of the influence of the inflated obstacle. This unexpected mapping will result in the loss of the path through the channel.

## Global path planning in a large-scale and complex multi-island environment

To verify the path planning ability and the computational efficiency improvement of the proposed algorithm in a large-scale and relatively complex multi-island environment, the simulation environment shown in Fig. 11B is selected, and the long-length path cases of USVs around the islands or across channels are investigated.

### Path planning cases

Five typical path cases were selected to demonstrate the performance of the proposed algorithm. The $P_s$ and $P_g$ groups are listed in Table 3. When determining the inshore-distance parameter $D_{sc}$, a USV with $v_{U,max} = 6$m/s, $t_r = 2$s, $a_d = -1$m/s² is considered as a case. Based on Eq. (20), $D_{sc}$ is suggested to select a value larger than 30 m. Therefore, a larger value $D_{sc} = 50$m is used for the path planning of the selected five path cases. Another

**Table 3  Quantified information of the five typical paths.**

| Path | $P_s$ (km) | $P_g$ (km) | Length (km) |
|------|-----------|-----------|-------------|
| $\ell_1$ | [35.34, 39.25] | [15.31, 11.65] | 38.84 |
| $\ell_2$ | [19.42, 41.02] | [17.10, 3.63] | 37.90 |
| $\ell_3$ | [42.96, 43.67] | [46.34, 8.24] | 38.20 |
| $\ell_4$ | [36.11, 18.77] | [47.44, 41.01] | 26.47 |
| $\ell_5$ | [3.95, 26.52] | [50.45, 30.83] | 47.82 |

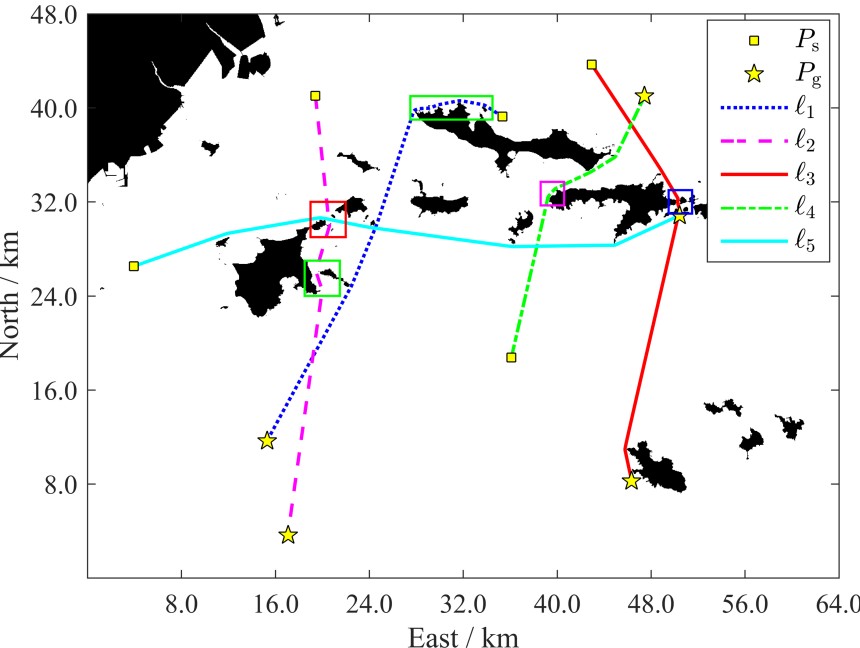

**Figure 14  Five typical paths in the surrounding area of Changhai County.** Several portions of these paths marked by rectangles are amplified to show more details in Fig. 15.

inshore-distance parameter $D_{Th} = 200$ m is selected empirically. The main parameters used in these path planning cases are shown in Table 1.

As shown in Fig. 14, all paths successfully bypass the islands. The enlarged views (see Fig. 15, every path is displayed with a solid line) provide further details. Every path maintains a relatively safe distance when close to an island, and smooth when turning around the island. For a narrow channel of a certain degree (usually wider than $2D_{LRes}$), the path is planned along the quasi-midline of the channel to ensure that it is as safety as possible. The path lengths range from about 26.52 km to 47.86 km (see Table 3). These lengths can cover the range of most applications of current small-and medium-sized USVs. These results show the effective path planning ability of the proposed algorithm in a large-scale and complex multi-island environment.

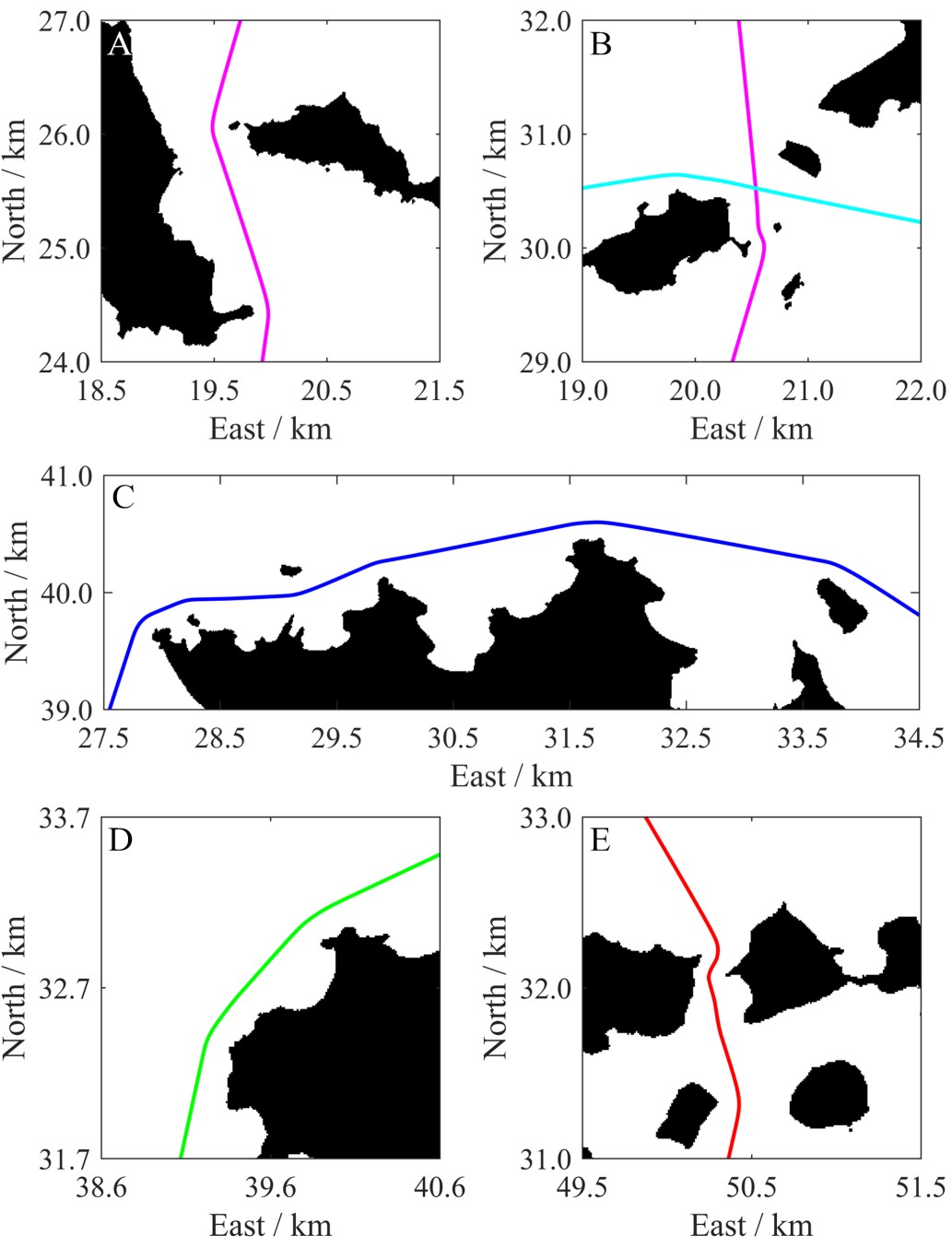

**Figure 15 Enlarged views of path portions.** (A)–(E) show the portions that are indicated by rectangles in Fig. 14. Detailed information can be acquired from (A) to (E). For example, all paths maintain a relatively safe distance from each island, the paths are smooth even when they turn around islands, and paths across narrow channels can be planned along the quasi-midline of the channels.

**Table 4** Planning times of five typical path cases for the compared algorithm based on a single HSR grid map and the proposed algorithm based on two-level SR grid maps.

| Path | Average planning times (s) | |
|---|---|---|
| | Compared algorithm | Proposed algorithm |
| $\ell_1$ | 29.21 | 1.94 |
| $\ell_2$ | 24.65 | 1.95 |
| $\ell_3$ | 30.25 | 1.97 |
| $\ell_4$ | 16.44 | 1.76 |
| $\ell_5$ | 40.40 | 2.24 |

### Computational efficiency improvement

To verify the computational efficiency improvement of the proposed algorithm, the time spent on the five typical paths (see Table 3) based on the proposed algorithm was calculated. For comparison, the time spent on the basic algorithm which is executed directly on a single HSR grid map by applying the IDC-FM$^2$ method was also calculated as a reference. The C algorithm code was tested on a computer with a Core i5-6300U CPU and 8G memory, which runs a 64-bit Win7 operating system.

Path planning for every typical path was repeated 10 times. Every planning time starts from the HSR grid map reading (in *Steps* S1 and T1) and ends when the global path has been calculated (in *Steps* S2 and T5). Because the path planning is performed on a Windows operating system, which involves multitasking, there may be many factors influencing the planning time, such as the variable CPU usage, random CPU hit rate to the cache, and possible thread scheduling. Therefore, the average planning time is used to evaluate the computational efficiency. The average planning time results of the five planned paths are presented in Table 4. These results indicate that the computational efficiency of the proposed algorithm based on two-level grid maps is significantly higher than the algorithm based on a single HSR grid map. The time for all cases is approximately 2 s, indicating that this method can be used in less demanding real-time planning applications. This can effectively improve the practicality of the proposed algorithm.

When planning a path by applying the basic FM$^2$-based algorithm, if the grid map scale is very large, two aspects of calculations will severely increase compared with small-scale maps. First, as the number of cells increases significantly, the calculations for free cells increase. Second, $\mathcal{S}_T$ changes in every iteration of calculations. The overall scale of $\mathcal{S}_T$ clearly increases, resulting in the increase in sorting time of the adopted priority heap struct. This compared algorithm, whose algorithm flow is shown in Fig. 8, is used directly on a single HSR grid map. The valid scale of the HSR grid map is the number of free cells. For comparison, the proposed algorithm determines the effective regions within the HSR grid map first and then plans a path within the determined effective regions. As shown in Fig. 9, *Steps* T1–T3 complete the determination of effective regions, and *Steps* T4–T5 realize the path planning. The same function of the path planning is also realized by *Steps* S1–S2, as shown in Fig. 8. Comparing the path planning steps of the compared and proposed algorithms, the numbers of free cells in effective regions are much less than the ones in the HSR grid map. This indicates that the calculations for free cells and the overall

scale of $\mathcal{S}_T$ will greatly decrease. Therefore, the planning time of the proposed algorithm reduces. Certainly, there are more steps in the proposed algorithm. Planning time spent on these steps (*Steps* T1–T3) mainly depends on the LSR mapping parameter $L$. When the planning time spent on the compared algorithm, $t_{ref}$, is large enough, the planning time spent on *Step* T2 can reduce to less than $t_{ref}/L^2$ (in the cases shown in Table 4, this value is approximately $t_{ref}/(2L^2)$). The planning time spent on *Steps* T1 and T3 is often much less than $t_{ref}$. Therefore, the increased planning time spent on *Steps* T1–T3 is much less than the saved planning time spent on *Steps* T4–T5.

## CONCLUSIONS

In some special USV applications such as sea rescue, it requires the USV to reach a goal position as soon as possible. FMM is a suitable global path planning method which can plan a time-optimal global path. However, the planned paths are not safe enough when they bypass obstacles, because of that they are too close to the obstacles. Therefore, FMM should be improved for safety considerations. One classical approach is applying the FMM within the map with inflated obstacles. Although the planned paths are safe by inflating the obstacle size, there may be abrupt turns when the obstacles have sharp corners, and the planned paths are influenced by shapes of inflated obstacles and may be not very smooth. Such paths are usually unfriendly for USVs. A rapid global path planning algorithm applying an IDC-FM$^2$ method in two-level SR grid maps is proposed for USV applications with short time requirements in large-scale and complex multi-island environments. This algorithm can acquire a continuous, smooth, quasi-time-optimal path while maintaining a safe distance around obstacles when bypassing obstacles. When the path is near obstacles, it is limited to a safe area determined by two inshore-distance parameters. By adjusting these two inshore-distance parameters, the path can be modified flexibly. Although the time optimality is missed by using the IDC-FM$^2$ method, the safety with a higher priority in most applications has been ensured initially while the optimal loss of time remains at a certain level. The two-path planning process based on two-level SR grid maps improves the computational efficiency compared with the basic FM$^2$-based method. The planning time on the order of seconds is acceptable in many global path planning applications of USVs. Meanwhile, the planning time of this order of magnitude is typically short enough for USVs in most situations with the requirement to replan the path. This indicates the potential of replanning by using the proposed algorithm from the perspective of planning time. As a comparison, when using the classical approach which applies the FMM within the map with inflated obstacles, narrow channels will be easier to map as obstacles when mapping the HSR grid map to the LSR grid map in the rapid planning process based on two-level SR grid maps. This shortcoming may result in the loss of some possible paths through channels.

On the other hand, there are still some challenges when using the proposed algorithm. For example, how to accurately obtain the location information of newly detected obstacles and how to add this information into the grid map when replanning are the common challenges. The IDC-FM$^2$ method also needs to be modified to plan the path meeting the

dynamic characteristics of a USV. Otherwise, the USV may not follow the replanned path successfully at the beginning of the path.

One shortcoming of the proposed algorithm is that paths through channels that are very narrow but still passable in reality may be missed because of the first path planning process in the LSR grid map. The introduction of environmental effects into the proposed algorithm is an important task to be performed in future works. Marine experiments should also be conducted to verify the validity of the algorithm.

## ACKNOWLEDGEMENTS

We would like to thank Editage for English language editing.

### Funding

This work was supported by the National Key R&D Program of China (No. 2017YFC14052**). The funders had no role in study design, data collection and analysis, decision to publish, or preparation of the manuscript.

### Grant Disclosures

The following grant information was disclosed by the authors:
National Key R&D Program of China: 2017YFC14052**.

### Competing Interests

The authors declare there are no competing interests.

### Author Contributions

- Dong Wang conceived and designed the experiments, performed the experiments, analyzed the data, performed the computation work, prepared figures and/or tables, authored or reviewed drafts of the paper, and approved the final draft.
- Jie Zhang, Deqing Liu and Xingpeng Mao analyzed the data, authored or reviewed drafts of the paper, and approved the final draft.
- Jiucai Jin conceived and designed the experiments, analyzed the data, authored or reviewed drafts of the paper, and approved the final draft.

### Data Availability

The raw data is available in the Supplemental Files.

### Supplemental Information

Supplemental information for this article can be found online at http://dx.doi.org/10.7717/peerj-cs.612#supplemental-information.

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
