# Peer review of "Rapid global path planning algorithm for unmanned surface vehicles in large-scale and multi-island marine environments"

_PeerJ Computer Science, doi:10.7717/peerj-cs.612_

## Round 0.1 · original submission · Major Revisions

This paper proposes a rapid global path planning algorithm for unmanned surface vehicles in large-scale and multi-island marine environments. The topic is interesting and relevant for robotics literature. However, reviewers' comments note that it is required to revise and resubmit it, considering the technical issues noted by the reviewers.

·

Basic reporting

The paper is well organized and is quite good from the English aspect. The introduction and background literature references are sufficient. Moreover, the shared Raw data supports the presented results. Finally, figures have good quality and tables are appropriately described.

Experimental design

The paper is into the scope of the journal. The contribution of the paper is clear, and it provides sufficient detail to replicate the proposed method.

Validity of the findings

The results of the paper show the effectiveness of the proposed approach. Moreover, a comparison against a basic approach is presented. The conclusions are well stated and connected to the original question investigated. However, a sentence of the conclusions is not clear. Comment to the author is:

1.- In the “Conclusions” section, the authors mention “Additionally, the execution time is sufficiently low in many global path planning applications of USVs”. Do you mean that the proposed algorithm execution time is low enough for global path planning applications? Please, discuss the common execution time for global path planning algorithms.

Additional comments

There are some minor problems that need attention. Comment and suggestion are:
1.- In Eq. (3), verify if there is an extra comma in T_(x_m x_i) > T_(x_m),T_(x_i).
2.- In Eq. (5), non-italic T is not defined.
3.- In line 179, verify the text “To The two-level SR grid maps are contained in the HSR and LSR grid maps.” I think “To” can be removed.
4.- In line 201, It seems to be missing the word “axis” for “X-“.
5.- In line 212, verify the text “This In the basic FM2 method,” I think “This” can be removed.
6.- Eq. (18) is hard to read because it is too small.

Some comments based on the “Instructions for Authors” of PeerJ are:
1.- In Fig. 10, the description of the Figures (A-B) should be included in the figure caption. Similarly, in Fig. 13, the description of the Figures (A-E) should be included in the figure caption.
2.- In Fig. 9, label each part with an uppercase letter. Then, the description of the Figures should be included in the caption.

Reviewer 2 ·

Basic reporting

The idea of the paper is interesting and relevant to the field. The introduction adequately describes the context and the related literature, in particular identifying some shortcomings of the FMM-based methods. However, in the last paragraph of the introduction, it must be clearly remarked the issues of the existing methods. It must be clarify if this is the first time that the IDC is introduced in an FMM method. At the beginning of the section “Related methods” it must be provided main references where the FMM method is detailed. The structure of the paper is good and the figures are relevant and of good quality. A remarkable issue is that the English writing can be improved and must be carefully revised.

Experimental design

The research reported in this work seems original, the research question is well defined and relevant. The proposed method is adequately described, although some aspects that will be described in the comments to the authors can be improved.

Validity of the findings

The results show the benefits of the proposed method in comparison to a basic FMM-based method in terms of flexibility to ensure a smooth and safe path. Some final questions about the computational efficiency must be clarified as will be specified in the comments to the authors.

Additional comments

The paper presents a global path planning algorithm for unmanned surface vehicles with time optimality requirements. The proposed method is based on the fast marching method, which has been improved to find better solutions in terms of safety and computational efficiency.

The idea of the paper is interesting and relevant to the field. The introduction adequately describes the context and the related literature, in particular identifying some shortcomings of the FMM-based methods. However, in the last paragraph of the introduction, it must be clearly remarked the issues of the existing methods. It must be clarify if this is the first time that the IDC is introduced in an FMM method. At the beginning of the section “Related methods” it must be provided main references where the FMM method is detailed. The structure of the paper is good and the figures are relevant and of good quality.

The research reported in this work seems original, the research question is well defined and relevant. The proposed method is adequately described, although some aspects must be better described or justified: 1) What is the importance to plan time-optimal paths in comparison to optimal paths in distance? At the end, the method modifies the paths according to the parameters of the IDC and time optimality is then missed. 2) It is not clear the effect of the vehicle dynamics (velocity, acceleration) over the final planned path. Equation (20) specifies a condition that the parameter Dsc must satisfy, but the results does not show how the dynamic parameters are taken into account in the method, since one expects that they are very important to find time-optimal paths. 3) Provide an interpretation of equation (1) besides of the definition of the variables. 4) Justify the need of the proposed method in comparison to the classical approach to inflate the obstacles size to guarantee to find collision free paths.

The results show the benefits of the proposed method in comparison to a basic FMM-based method in terms of flexibility to find a smooth and safe path. Regarding the reported results about computational efficiency, some issues must be addressed: 1) Clarify in the description of the proposed method, where exactly the method saves computational time with respect to the compared method. 2) Specify which stages of the proposed method are measured in the planning times reported in Table 4 (they must be called planning times instead of execution times). 3) It is not clear why these times varies from one run to another, clarify if there is some random process in the method that generates the difference in time. 4) Comment in the conclusions if the method can be extended to make replanning in execution time and what is the challenge to do it using this kind of methods.

Finally, a remarkable issue of the paper is that the English writing can be improved and must be carefully revised. For instance, 1) the second phrase of the abstract is not correct, 2) the word “grid” is incorrectly used most of the time, “cell” must be used in many cases. 3) In the paragraph starting in line 134, are the authors talking about the proposed method or the existing one? 4) Line 147, different behaviors… These kind of behaviors must be introduced to give a clearer notion. 5) Lines 179 and 212, among others.

---

## Round 0.2 · Minor Revisions

The paper has been improved since its last version. However, there are minor comments that need to be considered in a new version of this paper. Please take into account all the reviewers' comments for the new version of your paper.

·

Basic reporting

The paper is well organized and is quite good from the English aspect. The introduction and background literature references are sufficient. Moreover, the shared Raw data supports the presented results. Finally, figures have good quality and tables are appropriately described.

Experimental design

The paper is into the scope of the journal. The contribution of the paper is clear, and it provides sufficient detail to replicate the proposed method.

Validity of the findings

The results of the paper show the effectiveness of the proposed approach. Moreover, a comparison against a basic approach is presented. The conclusions are well stated and connected to the original question investigated.

Additional comments

The authors have followed the suggestions and comments to improve the manuscript quality. I have no further questions.

Reviewer 2 ·

Basic reporting

The paper have been adequately improved and I only suggest to include disscussions about two points that were replied to this reviewer but not included in the manuscript. This is specified in the comments to the authors.

Experimental design

This reviewer is satisfied with the experimental design presented in the paper.

Validity of the findings

This reviewer is convinced of the validity of the findings presented in the paper.

Additional comments

The paper have been adequately improved and I only suggest to include disscussions about two points that were replied to this reviewer but not included in the manuscript:

-What is the importance to plan time-optimal paths in comparison to optimal paths in distance? At the end, the method modifies the paths according to the parameters of the IDC and time optimality is then missed. Some discussion about this point must be included in the manuscript as was answered to the reviewer.

-Justify the need of the proposed method in comparison to the classical approach to inflate the obstacles size to guarantee to find collision free paths. Some discussion about this point must be included in the manuscript as was answered to the reviewer.

---

## Round 0.3 · accepted · Accept

Authors have been taken into account all the reviewers' comments; therefore, the paper can now be accepted for publication at PeerJ Computer Science